# Patterns of Response to Immune Checkpoint Inhibitors in Association with Genomic and Clinical Features in Patients with Head and Neck Squamous Cell Carcinoma (HNSCC)

**DOI:** 10.3390/cancers13020286

**Published:** 2021-01-14

**Authors:** Panagiota Economopoulou, Maria Anastasiou, George Papaxoinis, Nikolaos Spathas, Aris Spathis, Nikolaos Oikonomopoulos, Ioannis Kotsantis, Onoufrios Tsavaris, Maria Gkotzamanidou, Niki Gavrielatou, Elena Vagia, Efthymios Kyrodimos, Eleni Gagari, Evangelos Giotakis, Alexander Delides, Amanda Psyrri

**Affiliations:** 1Section of Medical Oncology, Second Department of Internal Medicine, National and Kapodistrian University of Athens, Attikon University Hospital, 12462 Athens, Greece; panagiota_oiko@hotmail.com (P.E.); miriamanastasiou9@gmail.com (M.A.); nikspa2011@gmail.com (N.S.); ikotsantis@gmail.com (I.K.); onoufriosmd79@yahoo.com (O.T.); mgkotzam@med.uoa.gr (M.G.); niki.gavrielatou@yale.edu (N.G.); elena.vagia@northwestern.edu (E.V.); 2Second Department of Medical Oncology, Agios Savas Anticancer Hospital, 11522 Athens, Greece; georgexoinis@gmail.com; 3Second Department of Pathology, National and Kapodistrian University of Athens, Attikon University Hospital, 12462 Athens, Greece; aspathis@med.uoa.gr (A.S.); noikonom1959@gmail.com (N.O.); 4Department of Otolaryngology-Head and Neck Surgery, Hippokration General Hospital, University of Athens, 11527 Athens, Greece; timkirodimos@hotmail.com; 5Oral Medicine Clinics, A. Syggros Hospital of Dermatologic and Venereal Diseases, Department of Dermatology, School of Medicine, University of Athens, 16121 Athens, Greece; egagari@med.uoa.gr; 6Department of Otorhinolaryngology, Facial Plastic and Reconstructive Surgery, Städtisches Klinikum Karlsruhe, 76133 Karlsruhe, Germany; egiotakis@med.uoa.gr; 7Second Otolaryngology Department, Attikon University Hospital, 12462 Athens, Greece; adelidis@med.uoa.gr

**Keywords:** hyperprogression, head and neck cancer, immunotherapy, TGK

## Abstract

**Simple Summary:**

Immunotherapy agents, such as immune checkpoint inhibitors (ICIs), act through different mechanisms compared to conventional chemotherapy and are characterized by unique patterns of response, such as hyperprogression (HPD), which refers to the paradoxical acceleration of tumor growth kinetics (TGK). In this regard, we sought to compare patterns of response to ICIs with respect to clinical and genomic features in a cohort of patients with recurrent/metastatic head and neck squamous cell carcinoma (HNSCC). In our cohort, HPD was observed in 15.4% of patients. We report for the first time an association of HPD with both shorter progression free survival and overall survival in HNSCC. Importantly, in a multivariate Cox analysis, the presence of HPD remained an independent prognostic factor for survival. Primary site in the oral cavity and administration of ICI in the second/third setting were significant predictors of HPD in multivariate analysis. Genomic profiling revealed that gene amplification was more common in HPD patients.

**Abstract:**

*Background*: We sought to compare patterns of response to immune checkpoint inhibitors (ICI) with respect to clinical and genomic features in a retrospective cohort of patients with recurrent/metastatic (R/M) head and neck squamous cell carcinoma (HNSCC). *Methods*: One hundred seventeen patients with R/M HNSCC treated with ICI were included in this study. Tumor growth kinetics (TGK) prior to and TGK upon immunotherapy (IO) was available for 49 patients. The TGK ratio (TGKR, the ratio of tumor growth velocity before and upon treatment) was calculated. Hyperprogression (HPD) was defined as TGKR ≥ 2. *Results*: HPD was documented in 18 patients (15.4% of the whole cohort). Patients with HPD had statistically significant shorter progression free survival (PFS) (median PFS 1.8 months (95% CI, 1.03–2.69) vs. 6.1 months for patients with non-HPD (95% CI, 4.78–7.47), *p* = 0.0001) and overall survival (OS) (median OS 6.53 months (95% CI, 0–13.39) vs. 15 months in patients with non HPD (95% CI, 7.1–22.8), *p* = 0.0018). In a multivariate Cox analysis, the presence of HPD remained an independent prognostic factor (*p* = 0.049). Primary site in the oral cavity and administration of ICI in the second/third setting were significant predictors of HPD in multivariate analysis (*p* = 0.028 and *p* = 0.012, respectively). Genomic profiling revealed that gene amplification was more common in HPD patients. *EGFR* gene amplification was only observed in HPD patients, but the number of events was inadequate for the analysis to reach statistical significance. The previously described *MDM2* amplification was not identified. *Conclusions:* HPD was observed in 15.4 % of patients with R/M HNSCC treated with IO and was associated with worse PFS and OS. *EGFR* amplification was identified in patients with HPD.

## 1. Introduction

The prognosis of patients with recurrent or metastatic (R/M) head and neck squamous cell carcinoma (HNSCC) is dismal, with the median survival ranging from 6 to 12 months depending on patient and disease-related factors [1].

We have recently become witnesses of a great clinical success achieved by immunotherapy in various cancers, including HNSCC [2]. Programmed Cell Death-1 (PD-1) is a transmembrane immune checkpoint that normally blocks T cell activation; anti-PD-1 monoclonal antibodies that abrogate PD-1 and enhance T cell effector function immune checkpoint inhibitors (ICIs) have been shown to improve overall survival (OS) in several tumor types [3]. In HNSCC, anti-PD-1 antibodies nivolumab and pembrolizumab have been both approved by the USA Food and Drug Administration (FDA) and European Medicine’s Agency (EMA) for platinum-refractory R/M disease based on landmark clinical trials demonstrating superior OS or durable response rate (RR), respectively, in patients treated with these agents [4,5]. More recently, based on the results of the phase III Keynote 048 clinical trial, pembrolizumab has been approved by both FDA and EMA in the first-line setting either as monotherapy or in combination with chemotherapy depending on tumor Programmed Cell Death Protein- Ligand-1 (PD-L1) expression status [6].

ICIs act through distinct mechanisms of action that ultimately differ from conventional treatments; additionally, they are characterized by unique side effects and patterns of response [7]. In this regard, there is growing evidence that immunotherapy may be harmful for a subset of patients who experience hyperprogression (HPD). HPD refers to the paradoxical acceleration of tumor growth kinetics (TGK), an approach that estimates the increase in tumor volume over time by integrating the time between imaging studies [8,9]. HPD has been reported across tumor types in 4%–29% of patients, and it is suggested that it might affect response to subsequent therapies [10].

The objectives of the present study were to compare the different patterns of response to immunotherapy and subsequently explore the prevalence, clinical impact, and genomic features associated with HPD in a retrospective cohort of patients with R/M HNSCC treated with ICIs.

## 2. Results

### 2.1. Patient Characteristics

One hundred and seventeen patients with R/M HNSCC were enrolled in the study. Three patients were excluded from the analysis because they were given a combination of ICI and chemotherapy. After collection and analysis of the radiological data, only 49 patients had a TGK ratio (TGKR) that was deemed evaluable for assessment, as they had imaging studies at three timepoints; prior, on, and after ICI administration (non-measurable tumors were excluded). The REMARK diagram of the study is shown in Appendix A. Baseline patient characteristics are demonstrated in Appendix A. The median age was 62 years (range: 40–88); the majority of patients were men and were younger than 65 years of age. Regarding tobacco consumption, most patients had a history of heavy smoking (*n* = 35, 74.5%), whereas alcohol abuse was reported by 14 patients (*n* = 30.4%). Finally, the most frequent primary tumor site was the oral cavity (*n* = 22, 44.9%).

### 2.2. Immunotherapy Delivery

The majority of patients received anti-PD-1 monoclonal antibody (*n* = 45, 91.8%), two (4.1%) received anti-PD-L1 and two (4.1%) received anti-PD-L1/anti-CTLA4 combination. The median number of cycles was six (range: 2–52). Immunotherapy was discontinued due to grade three arthritis in one patient. No other delay in administration or drug interruption/discontinuation was reported. Twenty-seven (55.1%) patients received immunotherapy as a first-line treatment (included in clinical trials with no PD-L1 status available), 20 (40.8%) as a second-line treatment, and two (4.1%) as a third-line treatment. Overall, PD-L1 Combined Positive Score (CPS) status was available in 14 patients; among them, three patients had PD-L1 CPS ≤ 1, six patients had PD-L1 CPS 1–19, and five patients had PD-L1 CPS ≥ 20. 

### 2.3. Response to Immunotherapy and Types of Progressive Disease

At a median follow-up of 10.7 months (range: 1.87–37.87) after immunotherapy initiation, 42 (85.7%) patients progressed, and 36 (73.5%) died. Median PFS was 2.8 months (95% CI, 2.6–5.8), while the median OS was 10.7 months (95% CI, 8.6–12.9). Overall response rate (ORR) to ICI was 20.4% (*n* = 10); two patients (4.1%) had complete response (CR), and eight patients (16.3%) had a partial response (PR). Stable disease (SD) was observed in 12 (24.5%) patients, while progressive disease (PD) as best response was observed in 27 (55.1%) patients. Among 42 patients that finally progressed, 18 patients fulfilled the criteria of HPD (42.8% of patients with PD, 15.4% of the whole cohort). Patients with HPD had a median TGKR of 3.18 (range: 2.08–43.03). 

### 2.4. Association of Clinical Characteristics to HPD

Patients with HPD were of younger age (median 54 vs. 65, *p* = 0.031, Mann–Whintey U), with primary site in the oral cavity (66.3% vs. 32.3% *p* = 0.036, Fisher’s exact test), receiving ICI as a second- or third-line of therapy (66.7% vs. 32.3%, *p* = 0.036). Results are summarized in Appendix A. 

Using a binary logistic model with HPD/non-HPD as the outcome, three parameters were significantly associated with the presence of HPD; age having a negative correlation (i.e., younger patients were more commonly HPD), the primary site being oral cavity (odds ratio (OR) 4.20, *p* = 0.023), and ICI treatment being the second- or third-line treatment (OR 4.20, *p* = 0.023). When we analyzed all tree parameters in a multivariate model, primary site (*p* = 0.028) and line of ICI (*p* = 0.012) remained significant predictors while age trended towards significance (*p* = 0.051). As shown previously, HPD correlated significantly with PD in less than three months from ICI start (OR 19.5, *p* < 0.001). 

In addition, a positive PD-L1 CPS score (≥1) was significantly associated with HPD (*p* = 0.011).

### 2.5. Association of Type of Progressive Disease with Survival

Patients with HPD had statistically significant shorter PFS compared to those with non-HPD (median PFS 1.8 months (95% CI, 1.03–2.69) vs. 6.1 months, respectively (95% CI, 4.78–7.47), *p* = 0.0001) (Figure 1). In addition, a statistically significant decrease in OS was observed in patients with HPD as compared to those with non-HPD (median OS 6.53 months (95% CI, 0–13.39) vs. 15 months respectively (95% CI, 7.1–22.8), *p* = 0.0018) (Figure 2). Univariate Cox proportional hazard model indicated the presence of HPD, local PD, and Eastern Cooperative Oncology Group (ECOG) performance status as significant prognostic factors for OS (Table 1). In a multivariate model, all three parameters remained significant. The presence of HPD remained an independent prognostic factor (*p* = 0.049) after checking for age, sex, smoking, alcohol, line of ICI therapy, and age groups.

Patients with HPD had statistically significant shorter PFS compared to those with non-HPD (median PFS 1.8 months (95% CI, 1.03–2.69) vs. 6.1 months respectively (95% CI, 4.78–7.47), *p* = 0.0001).

Patients with HPD had statistically significant decreased OS compared to those with non-HPD (median OS 6.53 months (95% CI, 0–13.39) vs.15 months respectively (95% CI, 7.1–22.8), *p* = 0.0018).

When using the second method for evaluation of tumor growth (ΔTGR) as previously proposed [11], we found a statistically difference in PFS between hyperprogressors and non-hyperprogressors (median PFS 1.8 months (95% CI, 1.51–2.08) vs. 3.1 months (95% CI, 1.29–4.90) respectively, *p* = 0.002), but no difference in OS (median OS 11.06 (95 % CI, 0–22.58) in patients with HPD vs. 12.53 months (95% CI, 8.27–16.78) in patients with non-HPD, *p* = 0.674). 

Regarding post-PD survival, there was no significant difference between patients with HPD as compared to patients with non-HPD (median 4.83 months (95% CI, 3.61–6.04) vs. 7.30 months (95% CI, 1.09–13.51), *p* = 0.872). Respective survival curves are demonstrated in Figure 3.

There was no significant difference in post-PD survival between patients with HPD as compared to patients with non-HPD (median 4.83 months (95% CI, 3.61–6.04) vs. 7.30 months (95% CI, 1.09–13.51), *p* = 0.872).

### 2.6. Efficacy of Chemotherapy in Patients Who Progressed on Immunotherapy

Chemotherapy was administered to 33 patients (67.34%) who had progressed during immunotherapy. Evaluable disease response was available in 29 patients, 12 with HPD (41.4%) and 17 with non-HPD (58.6%). Among them, one patient had a CR (3.4%), seven had a PR (24.1%), three had SD (10.32%), and six experienced PD (20.64%), while 12 patients died without evaluation of disease. Among eight responders, four had a history of HPD. 

Median PFS with chemotherapy after immunotherapy failure was 4.1 months for patients with HPD (95% CI, 3.08–5.11) and 3.4 months for patients with non-HPD (95% CI 2.88–3.91) (*p* = 0.403) (Figure 4). Median OS with chemotherapy after immunotherapy was 4.1 months (95% CI, 2.68–5.51) for patients with HPD as compared to 6.36 months (95% CI, 1.44–11.29) for those with non HPD (*p* = 0.933) (Figure 5).

Median PFS with chemotherapy after immunotherapy failure was 4.1 months for patients with HPD (95% CI, 3.08–5.11) and 3.4 months for patients with non-HPD (95% CI, 2.88–3.91) (*p* = 0.403).

Median OS with chemotherapy after immunotherapy was 4.1 months (95% CI, 2.68–5.51) for patients with HPD as compared 6.36 months (95% CI, 1.44–11.29) for those with non HPD (*p* = 0.933).

### 2.7. Genomic Alterations in Hyperprogressors

Genomic DNA from baseline biopsies of tumors was purified and subjected to targeted sequencing for 44 patients. Among them, 19 patients were included in the analysis (i.e., had available TGK before and upon ICI); eight patients had HPD, and their results were compared to 11 non-HPD patients. Since the data were complex, we summarized the analysis in gene mutations found more than one time in each category, and we tried to group them according to their biological action. The results are presented in Table 2.

Gene amplification was more frequent in HPD patients; in addition, *EGFR* gene amplification was only present in HPD patients, but the number of events was not adequate for the analysis to reach statistical significance. Gene amplification in HPD was identified for *EGFR*, *CCND1,* and *KRAS* while in non-HPD patients in *ERBB2* and *CCND1*.

Herein, we describe the mutational profiles of hyperprogressors:

Patient #1. Tumor DNA analysis revealed: (a) a *PIK3CA* E545K mutation. This mutation results in constitutive activation of PI3K/Akt/mTOR pathway [12], (b) a stop codon (p.R259Gfs*2) that inactivates the cytochrome P450 enzyme CYP2D6, which is primarily involved in drug metabolism, and (c) a *TP53* stop codon (p.Q104*) that results in protein loss (d) a genomic amplification of *EGFR* (37 copies). *EGFR* gene amplification is a frequent alteration in HNSCC and has been associated with increased EGFR oncogenic signaling [13]. 

Patient #2. Analysis of tumor DNA identified a *TP53* stop codon (p.R333Vfs*12) which results in protein loss.

Patient #3. Tumor DNA analysis revealed a p.Q1978* mutation in *NOTCH1* gene that introduces a premature stop codon, resulting in a truncated protein. The inactivation of the Notch signaling pathway due to loss of function mutations is frequent in HNSCC, highlighting the tumor suppressor role of Notch in this type of cancer [14].

Patient # 4. No mutations were identified in tumor DNA.

Patient #5. Analysis of tumor DNA identified a genomic *CCND1* amplification. If the amplification of *CCND1* is associated with overexpression of the corresponding protein, it might result in upregulation of the cell cycle and tumor aggressiveness [15]. 

Patient #6. Tumor DNA analysis revealed (a) an *EPHA2* G391R variant, which causes continuous activation of EPHA2, with increased phosphorylation of Src, cortactin, and p130. Activation of EpHA2 might cause invasiveness and migration of cancer cells [16]. (b) a *KNSTRN* pS24F variant, which has been associated with disruption of chromosomes and genetic instability [17] (c) a *NOTCH2* stop codon (pC783_K784delins*E), which results in loss of function of protein NOTCH2 and (d) a TP53 pE285K variant that results in protein loss.

Patient #7. Tumor DNA analysis identified (a) a stop codon leading to loss of function of protein NOTCH1 (p.Q290*) and (b) a genomic amplification of *KRAS,* which results in overexpression of KRAS protein, activation of KRAS-MAPK pathway, and tumor progression [18].

Patient #8. Tumor DNA analysis revealed (a) a genomic amplification of *IL-6* with an unknown effect on protein function, (b) a genomic amplification of *EGFR*, (c) a stop codon leading to a possible loss of function of protein KT2MD (p.R2645*), and (d) a V173L *TP53* variant associated with loss of function of the corresponding protein.

## 3. Discussion

The incorporation of immunotherapy in the treatment of several solid tumors and the widespread use of ICIs in daily oncology practice has put in the spotlight differences in the mechanism of action and novel patterns of response associated with those agents. Therefore, remarkable durability of response, delayed responses, initial disease progression that does not reflect treatment failure, as well as atypical patterns of response such as pseudoprogression and HPD are unique characteristics of ICIs that have been recently introduced. The concept of HPD, defined as the acceleration of disease that produces a harmful effect in a subset of patients treated with ICIs, has been initially reported in retrospective analyses that included patients treated with PD-1/PD-L1 inhibitors in phase I studies [19,20]. In our study, HPD was observed in 15.4% of patients with R/M HNSCC treated with PD-1/PD-L1 inhibitors. Importantly, the presence of HPD significantly correlated with shorter PFS (*p* = 0.0001) and OS (*p* = 0.0018).

In our series, the rate of HPD was 15.4%, which is lower than the one described in a previous report by Saada-Bouzid that included only patients with R/M HNSCC and identified a HPD rate of 29% (10 out of 34 patients) [20]. This inconsistency might reflect the difference in the number of patients included in the two studies (34 vs. 49 patients). Since the incorporation of ICIs in everyday clinical practice, several groups have reported on the incidence and clinical impact of HPD [8,19,21,22,23]. In the retrospective study by Champiat et al., HPD was observed in 9% of 131 patients with malignant solid tumors treated with immunotherapy [19]. This study included six patients with HNSCC; none of them had HPD. Kato et al., using a stricter definition of HPD, identified HPD in only six of 155 patients with R/M cancer; none of the 11 patients with HNSCC had HPD [21]. More recently, Ferrara et al. identified 56 hyperprogressors among 406 patients with NSCLC treated with immunotherapy [8]. Kanjanapan et al. identified a HPD rate of 7% in patients with solid tumors who participated in phase I studies; in this study, the most frequent tumor type was indeed HNSCC, and HPD was observed in two patients, one with laryngeal and one with oral cavity carcinoma [8]. Lastly, in a report by Matos et al., HPD was observed in 33 of 214 (15.4%) patients included in phase I trials [22].

Importantly, we report for the first time a statistically significant correlation of HPD with worse survival in patients with R/M HNSCC. In the report by Saada-Bouzid, a statistically significant difference in PFS was shown between patients with HPD and patients with non-HPD irrespective of the use of RECIST or iRECIST criteria (using RECIST criteria, 2.5 vs. 3.4 months respectively, *p* = 0.003; using iRECIST, 2.9 vs. 5.1 months respectively, *p* = 0.02), but no significant difference in OS was observed; in our study, we demonstrated a significant association of HPD with both PFS and OS [20]. Similarly, Ferrara et al. reported a correlation of HPD with shorter PFS in patients with non-small cell lung cancer (NSCLC) [8]. Notably, HPD has been previously related to poor survival when incorporating HPD into RECIST criteria subgroups (HPD vs. CR/PR) and not when comparing hyperprogressors to non-hyperprogressors [19].

A major concern that could also interpret the discrepancy in results between published studies is the various methodologies in HPD evaluation, as there is no consensus on the optimal definition [24]. Indeed, in HPD, ICIs provoke tumor growth through a distinctive effect, and evaluation of this paradoxical accelerated tumor progression is based on parameters that correlate with pretreatment tumor kinetics and early evolution after immunotherapy initiation [25]. For example, in the studies by Champiat et al. and Kanjanapan et al., the definition of HPD was based on Tumor Growth Rate (TGR), which evaluates the increase in tumor volume over time by taking into account two CT scans measurements [19,23]. In these two studies, hyperprogressors were patients that had PD by RECIST criteria at first evaluation and a greater than a two-fold increase of TGR before and during immunotherapy. Ferrara et al. used a similar definition, comparing pre- and post-immunotherapy TGR and defining HPD as PD by RECIST criteria at first evaluation and a difference in TGR higher than 50% [8]. On the other hand, Kato et al. and Matos et al. used a mixture of clinical and radiological criteria by incorporating TTF < 2 months in HPD definition [21,22]. In an attempt to comprehensively compare and unify different definitions of HPD, Kas et al. analyzed all methodologies of previous HPD definitions and applied them to a cohort of patients with NSCLC [11]. Interestingly, the authors observed several differences in the rate of HPD depending on the method used; in addition, they showed that only a minority of patients (4.7%) were covered by all definitions, suggesting that each methodology includes a distinct subgroup of patients with diverse patterns of tumor progression [26]. The authors suggested a new definition of HPD based on the difference between TGR before and during immunotherapy (ΔTGR) that was found to best correlate with poor OS. A ΔTGR of more than 100 was indicated to distinguish between patients with HPD and patients with PD that was not classified as HPD [11]. When this definition of HPD was applied in our cohort, no significant difference in survival was found; however, the study by Kas et al. and previous studies included patients with NSCLC and other solid tumors, and this definition might not reflect the complex tumor biology and growth kinetics of HNSCC.

The requirement of at least one radiological criterion in all studies excludes patients that have disease-related rapid clinical deterioration post immunotherapy, that is either not prominent on CT scans, or CT scans are unavailable due to forthcoming urgency in changing treatment; thus, the frequency of HPD could be underestimated. Indeed, Champiat and Ferrara comment on this controversy by noting that some patients included in their studies (8% and 30.5%, respectively) were unable to undergo CT scans due to rapid clinical deterioration and could not meet the required radiological criteria for HPD [8,19]. Lo Russo G et al. also defined HPD using a combination of clinical and radiological criteria [27]. In our study, we used the definition of radiological HPD described by Saada-Bouzid [20], based on the rationale that this study included only patients with HNSCC. However, we observed that several patients experienced clinical HPD without meeting radiological criteria; this indicates that HNSCC might possess distinct characteristics that complicate the assessment of response based solely on imaging criteria and raises the question of the incorporation of clinical benefit endpoints into the definition of response to immunotherapy in HNSCC.

A critical and clinically relevant question that emerges is whether HPD can be predicted by clinicopathological or genomic features. We found a significant correlation between HPD and younger age, primary site in the oral cavity, administration of immunotherapy at a later line (second or third), and PD in less than three months from initiation of ICI. We also found a significant correlation of HPD with PD-L1 CPS score, although definite conclusions cannot be drawn due to the limited availability of PD-L1 status’. Despite the meticulous study of clinical parameters, limited correlations to HPD have been reported. Several important clinicopathological factors, such as tumor histology, tumor burden, PD-L1 status, and previous treatment, have not been shown to be independent predictors of HPD in multivariate analyses [25]. In the paper by Saada-Bouzid, HPD was associated with regional recurrence in a previously irradiated field [20], emphasizing the role of radiation as an immune system modulator. Contrary to our finding, HPD has been linked to older age in the study by Champiat et al., but only in univariate analysis [19]. The association of HPD with both administration of immunotherapy at a later line and PD in less than three months from immunotherapy initiation might be related to tumor aggressiveness. Although a correlation of HPD with aggressive features, such as tumor burden, has not been proven significant [19], metastasis at more than two sites has been shown to independently predict HPD in NSCLC [8]. Regarding the association of PD-L1 CPS score with HPD, it is important to highlight a possible effect of the anti-PD-1 antibody on tumor cells. Anti-PD-1 antibody blocks the interaction of PD-1 with PD-L1 in responders. However, if the tumors also express PD-1, which signals an anti-tumor effect, it can be hypothesized that an anti-PD-1 antibody can accelerate tumor growth. PD-1 expression on tumor cells also has also been reported to be a reason for HPD in NSCLC. This observation is in agreement with the finding that PD-1 expression on NSCLC cells reduces tumor cell viability, and PD-1 blockade promotes tumor proliferation [28]. This suggests that anti-PD-1 therapy may not be beneficial or even deleterious to certain lung cancer patients. Thus, the assessment of intrinsic PD-1 marker expression on tumor biopsy might be needed before assigning the anti-PD-1 therapy to patients.

Subsequently, we sought to determine genomic biomarkers of HPD, by performing genomic profiling of tumor DNA in our cohort. Among 19 patients, eight were hyperprogressors. Contrary to the findings by previous reports [21,29], we found no *MDM2* family amplifications. Of note, Cowzer et al. also failed to identify *MDM2* amplifications among hyperprogressors in a lung cancer cohort [30]. Similar to Kato et al. [21], two hyperprogressors in our cohort had *EGFR* alterations, and when we compared genomic features of patients with HPD to 11 patients with non-HPD we found that this was a unique genomic feature of HPD patients. Overall, gene amplifications were found more commonly in HPD patients; however, the small number of events precluded statistical significance. Indeed, one patient with HPD had *KRAS* amplification, which has been previously reported in patients HPD in a cohort of patients with gastric cancer [31]. In addition, one patient had *CCND1* amplification, which has been shown to mediate immunosuppression and has been determined as an independent prognosticator for poor outcome and limited response to ICIs in HNSCC and other solid tumors [32,33]. On the other hand, three hyper progressors had mutations in *Notch* genes. It is known that Notch signaling regulates many components in the tumor microenvironment, such as the expression of pro-inflammatory cytokines IL1β and CCL2, and plays an important role in effector T cell differentiation [34,35]. Moreover, Notch signaling promotes the accumulation of myeloid-derived suppressor cells (MDSCs), which are major immune response regulators [36]. Notably, mutations in TP53 and PI3K genes did not differ between the two groups. Although these are important observations, these mechanisms of immune resistance must be assessed in the context of HPD; furthermore, the molecular profiles of hyperprogressors must be validated in larger cohorts.

Our study had several limitations. First, the absence of a control arm does not allow drawing definite conclusions about the nature of HPD and its association with ICIs. In the study by Ferrara et al., the authors used a control arm to enable comparison in response patterns between immunotherapy and chemotherapy and found a HPD rate of 5% in patients treated with chemotherapy [8]. This study suggests that HPD might not be a phenomenon uniquely observed in treatment with ICIs; indeed, rapid acceleration of clinical picture has been previously described in cancer patients, such as in the context of discontinuation of tyrosine kinase inhibitors [37]. A better understanding of the nature of HPD and identification of HPD predictors would require better knowledge of the natural history of cancer without treatment, which is currently difficult to pursue due to ethical issues. Second, the selection of target lesions for radiological evaluation might not be representative, especially in HNSCC. Lastly, the sample size in our study is too small to draw definite conclusions.

## 4. Materials and Methods 

### 4.1. Patient Selection

In this single-center study, data were retrospectively collected from all consecutive eligible patients with R/M HNSSC treated with immune checkpoint inhibitors (PD-1/PD-L1 inhibitors, PD-L1 plus anti-CTLA-4) from 27 October 2015 to 27 July 2020, at Attikon University Hospital, Athens, Greece. The present study was approved by the institutional review board of Attikon University hospital (Athens, Greece).

Eligible patients had to be 18 years or older, with histologically confirmed R/M squamous cell carcinoma of the oral cavity, oropharynx, hypopharynx, larynx, or nasal/cavity. Stage was determined according to the Tumor Node Metastasis (TNM) classification of the American Joint Committee for Cancer staging 8th edition.

### 4.2. Radiological Evaluation

Minimum imaging exams included CT scans of the head and neck, and thorax. CT scans at two different time points before immunotherapy initiation (baseline and pre-baseline) and one during treatment was mandatory for radiological evaluation. The pre-baseline CT-scan was required in order to evaluate TGK before starting immunotherapy. All patients were evaluated with CT scan during treatment at three month intervals starting from the beginning of immunotherapy. 

All scans were retrospectively reviewed. For each patient, target lesions were assessed using RECIST 1.1 [38] and RECIST-based immune criteria (irRECIST) [39]. Objective response rate (ORR), progression free survival (PFS), as well as TGK prior to and upon immunotherapy were evaluated.

The following data were obtained from each patient: age, sex, tobacco and alcohol consumption, primary tumor location, histology grade, Human Papillomavirus (HPV) status, TNM stage at diagnosis, ECOG performance status (PS), level of LDH, number of metastatic sites, presence of liver metastases, number, and type of previous and subsequent treatments, response to immunotherapy, response to subsequent treatments, pattern of progression to ICIs (either local or systematic), dates of imaging, days from the end of radiotherapy and the largest diameter of target lesions.

### 4.3. Tumor Growth Kinetics and Definition of HPD

To estimate HPD, we compared TGK prior and TGK upon immunotherapy. Tpre, T0, and Tpost refer to the time of pre-baseline, baseline, and first imaging, respectively. Spre, S0, and Spost refer to the sum of the largest diameter of target lesions at pre-baseline, baseline, and first imaging, respectively. We calculated TGKpre and TGKpost as previously described [20]. TGKpre was defined as the difference of the sum of the largest diameters of the target lesions per unit of time between pre-baseline and baseline imaging: (S0-Spre)/(T0-Tpre). Similarly, TGKpost was defined as: (Spost-S0)/(Tpost-T0). 

The TGK ratio (TGKR), which reflects tumor growth velocity, was calculated before and upon treatment. The TGKR was defined as the ratio of TGKpost to TGKpre. TGKR >1 indicated tumor growth acceleration, while 0 < TGKR < 1 and TGKR <0 indicated tumor growth deceleration and tumor shrinkage, respectively. HPD was defined as the presence of TGKR ≥ 2. A second method for the evaluation of tumor growth was used as previously described [11]. ΔTGR was calculated and the proposed cut-off of 100 was used.

### 4.4. Comprehensive Genomic Profiling 

Patients who agreed to provide informed consent had biopsies of tumor lesions. Using OncoDEEP (OncoDNA) sequencing panel, we performed targeted mutational analysis of baseline tumor DNA for 313 cancer-associated genes.

### 4.5. Statistical Analysis

Statistical comparisons were performed using χ^2^ or Fisher’s exact tests for categorical data and Mann–Whitney’s U test for continuous variables. To identify significant predictors of HPD, a binary logistic regression analysis was used for each parameter and all significant predictors were analyzed in a multivariate model.

Survival analysis included the construction of Kaplan-Meier survival curves, and the log-rank statistical test was performed for the comparison of median survival times and the estimation of the *p*-value. PFS-immuno was defined as the time from the date of immunotherapy initiation to the date of tumor progression (PD) or death from other causes or censored at the time of last contact. OS was defined as the time from the date of immunotherapy initiation to the date of death from any cause or censored at the time of last contact. Post-progression (OS-postPD) survival was defined as the time from the date of PD to immunotherapy to death from any cause or censored at the time of last contact. PFS-chemo was defined as the time from the date of chemotherapy initiation to the date of PD or death from other causes or censored at the time of last contact. OS-chemo was defined as the time from the date of chemotherapy initiation to the date of death from any cause or censored at the time of last contact. Survival time definitions are graphically described in Appendix A. For multivariate analysis of OS, a Cox proportional hazard risk model was utilized. 

Statistical analyses were performed using the SPSS Statistics v25.0 (IBM Corp., Armonk, NY, USA) software. Two-tailed tests were used, and *p*-values < 0.05 were considered statistically significant. 

## 5. Conclusions

In conclusion, we found a 15.4% incidence of HPD in a cohort of patients with R/M HNSCC treated with ICIs. We report for the first time a significant association of HPD with both PFS and OS in patients with R/M HNSCC. In our cohort, several clinical parameters such as younger age, primary site in the oral cavity, PD in less than three months, and treatment with ICIs in the second- and third-line settings correlated with the presence of HPD. In addition, molecular profiling of hyperprogressors revealed that gene amplifications were more common in patients with HPD; moreover, EGFR alterations were present only in HPD patients. Future studies should focus on molecular biomarkers with the view to optimize patient clinical outcomes.

## Figures and Tables

**Figure 1 cancers-13-00286-f001:**
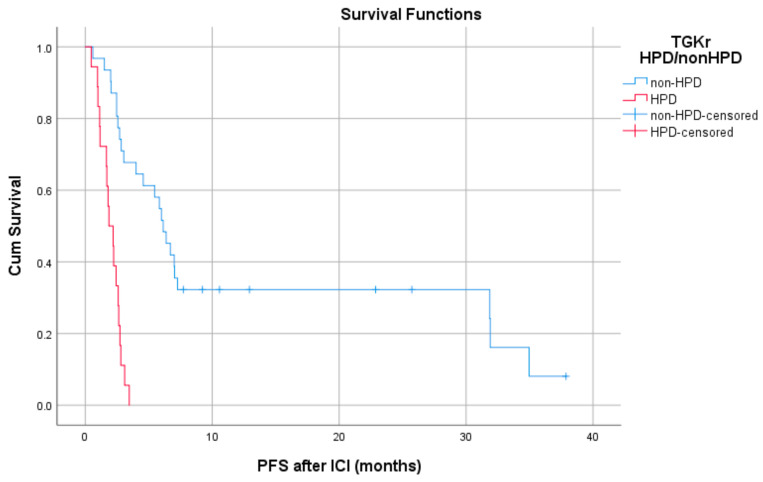
Association of type of progressive disease (hyperprogression (HPD) vs. non-HPD) with progression free survival (PFS) post immunotherapy initiation (ICI).

**Figure 2 cancers-13-00286-f002:**
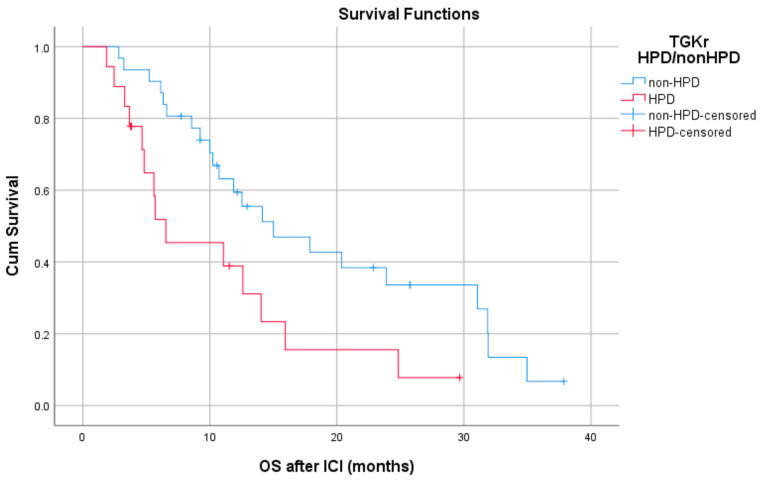
Association of type of progressive disease (HPD vs. non-HPD) with overall survival (OS) following immunotherapy initiation (ICI).

**Figure 3 cancers-13-00286-f003:**
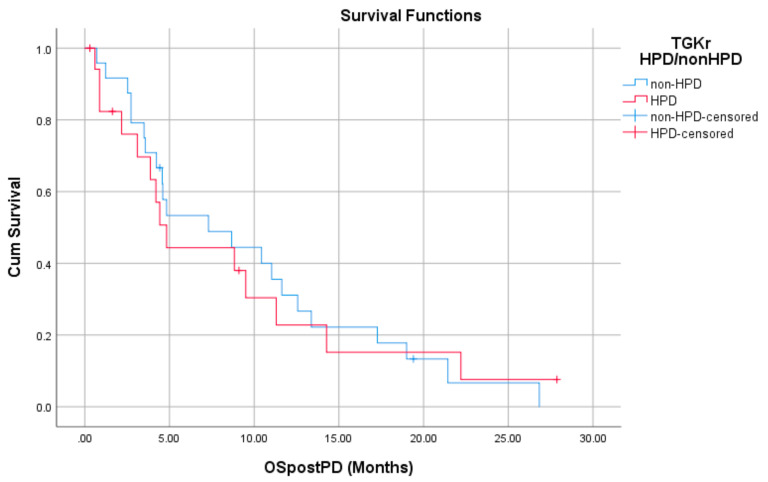
Association of type of progressive disease (PD) (HPD vs. non-HPD) with post-PD survival.

**Figure 4 cancers-13-00286-f004:**
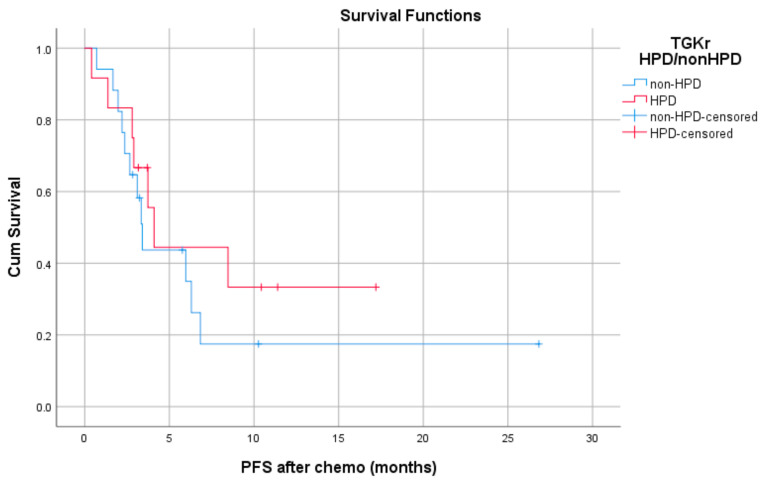
Association of type of progressive disease (HPD vs. non-HPD) with PFS of chemotherapy after immunotherapy failure (PFS chemo).

**Figure 5 cancers-13-00286-f005:**
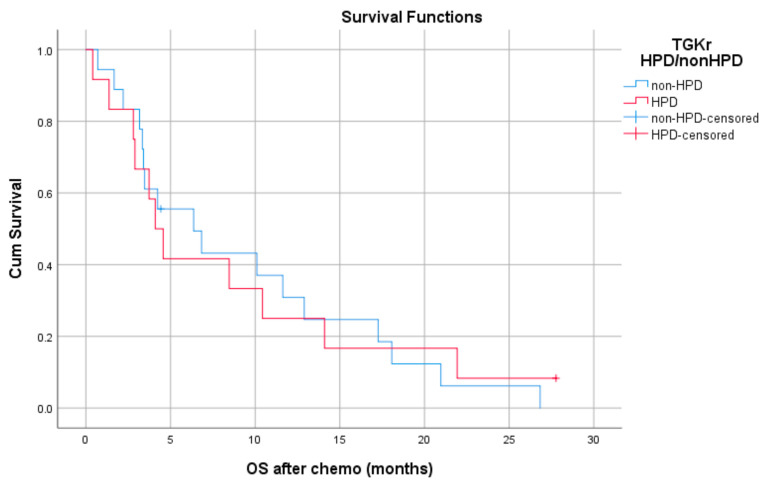
Association of type of progressive disease (HPD vs. non-HPD) with OS of chemotherapy after immunotherapy failure (OS chemo).

**Table 1 cancers-13-00286-t001:** Univariate cox proportional risk analysis for OS.

Co-Variates	Values	HR	95% CI	*p*-Value
Age (years)	≥65 vs. <65	1.02	0.51–2.03	0.951
Sex	Female vs. Male	1.24	0.81–1.89	0.313
Setting	Recurrent vs. metastatic	1.36	0.70–2.65	0.361
Alcohol consumption	Heavy vs. Light/No	0.85	0.58–1.24	0.413
Smoking	Heavy vs. Light/No	1.02	0.45–2.30	0.955
Primary site	Oral cavity vs. other	1.24	0.87–1.76	0.225
Line of immunotherapy	2^nd+^ vs. first	1.85	0.92–3.71	0.084
Type of PD (TGKR)	HPD vs. PD	1.72	0.85–3.48	0.131
	HPD vs. non-HPD	2.29	1.13–4.65	**0.021**
Type of PD (ΔTGR)	HPD vs. non-HPD	1.25	0.43–3.59	0.675
Recurrence (Local)	Present vs. Absent	1.96	0.87–4.03	0.065
Recurrence (Regional)	Present vs. Absent	0.86	0.44–1.65	0.645
Recurrence (Distant)	Present vs. Absent	0.59	0.28–1.23	0.163
Local PD	Present vs. Absent	2.43	1.18–4.97	**0.015**
Systematic PD	Present vs. Absent	1.41	0.65–3.04	0.380
LDH	Abnormal vs. normal	1.090	0.55–2.16	0.805
ECOG PS	1 vs. 0	2.56		**0.011**
	2 vs. 0	4.26		**0.009**

HR: hazard ratio; PD: progressive disease; HPD: hyperprogressive disease; TGKR: tumor growth kinetics ratio; ΔTGR: change in tumor growth rate; LDH: lactate dehydrogenase; ECOG PS: Eastern Cooperative Oncology group performance status. Statistical significance is noted in bold.

**Table 2 cancers-13-00286-t002:** Genomic alterations in hyperprogressors vs. in non hyperprogressors.

Gene	HPD Mutated	Non-HPD Mutated	*p*-Value
*TP53*	4 (50%)	7 (63.6%)	0.658
*PI3K*/*PTEN*/*FAT1*	2 (25%)	3 (27.3%)	1.000
*RAS*/*EGFR*	1 (12.5%)	2 (18.5%)	1.000
No mutation	1 (12.5%)	3 (27.3%)	0.603
Gene amplification	4 (50%)	2 (18.3%)	0.319
*EGFR* amplification	2 (25%)	0 (0%)	0.164
**Categories**	**HPD Mutated**	**Non-HPD Mutated**	***p*-Value**
Growth factors	4 (50%)	6 (54.5%)	1.000
DNA damage errors	5 (62.5%)	5 (62.5%)	0.650
Transcription factors Epigenetic	2 (25%)	3 (27.3%)	1.000
Ubiquitin Proteasome	0 (0%)	1 (9.1%)	1.000

## Data Availability

Data is contained within the article or Appendix A.

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
