# Peer review of "Patterns of Response to Immune Checkpoint Inhibitors in Association with Genomic and Clinical Features in Patients with Head and Neck Squamous Cell Carcinoma (HNSCC)"

_cancers, 2021, doi:10.3390/cancers13020286_

Round 1
Reviewer 1 Report
The manuscript reports the HPD in HNSCC patients treated with ICI. The salient features of the study are: retrospective analysis of the tumor growth kinetics (TGK) using CT scans data before, during and post ICI Rx revealed that about 15% of the patients are HPD and mostly in younger patients who are on ICI as 2nd or 3rd line of treatment. There were some gene alterations such as amplifications but sample size is not large enough to conclude indeed it is the cause of HPD. In the absence of histological data except for PD-L1 expression, it is important to discuss the possible effect of anti-PD-1 antibody on the tumor cells. Anti-PD-1 antibody works by blocking the interaction of PD-1 on anti-tumor T cells with PD-L1 in responders. However, if the tumors also express PD-1 which signals anti-tumor effect then anti-PD-1 antibody can accelerate tumor gorwth as suggested by the following study.
PD-1 expression on tumor cells also was reported to be a reason for HPD in NSCLC. This observation is in agreement with the finding of PD-1 expression by NSCLC cells reduces tumor cell viability and PD-1 blockade promotes tumor proliferation (Du S et al, Oncoimmunology 2018). This suggests that anti-PD-1 therapy may not be beneficial, or even deleterious to certain lung cancer patients. Thus, the assessment of intrinsic PD-1 marker expression on tumor biopsy is needed before assigning the anti-PD-1 therapy to patients.
Author Response
We thank the reviewer for his/her comments.
In the revised maunscript, a possible effect of the anti-PD-1 antibody on tumor cells is discussed in lines 297-306.
Reviewer 2 Report
The manuscript entitled “Patterns of response to immune checkpoint inhibitors 3 in association with genomic and clinical features in 4 patients with head and neck squamous cell 5 carcinoma (HNSCC)” deals with a very important topic that is understudied in clinical practice. To clarify the reasons leading to hyperprogression in different settings is of paramount importance. The authors shed light in this issue focusing on patients with Head and neck carcinoma. Overall, this study is interesting, and is expected to be of interest to readers of Cancers. There are few points that need to be addressed prior publication.
First the authors included in the title that genomic features are associated with hyperprogression, which stresses out the significance of this relationship. However, in the abstract this association is not mentioned at all.
Second, the authors found that gene amplification is more frequently observed in patients with hyperprogression. Could the authors provide more details about the genes that are amplified in this setting?
Author Response
We thank the reviewer for his/her comments.
In the revised manuscript, the abstract has been modified and association of genomic features with HPD is mentioned in lines 38-44.
Gene amplification in HPD was idenitified in EGFR, CCND1 and KRAS genes. This is mentioned in lines 180-181.